# Unleashing Spinal Cord Repair: The Role of cAMP-Specific PDE Inhibition in Attenuating Neuroinflammation and Boosting Regeneration after Traumatic Spinal Cord Injury

**DOI:** 10.3390/ijms24098135

**Published:** 2023-05-02

**Authors:** Femke Mussen, Jana Van Broeckhoven, Niels Hellings, Melissa Schepers, Tim Vanmierlo

**Affiliations:** 1Department of Neuroscience, Biomedical Research Institute BIOMED, Hasselt University, 3590 Diepenbeek, Belgium; 2University MS Center (UMSC) Hasselt-Pelt, Hasselt University, 3500 Hasselt, Belgium; 3Department of Immunology and Infection, Biomedical Research Institute BIOMED, Hasselt University, 3590 Diepenbeek, Belgium; 4Department of Psychiatry and Neuropsychology, School for Mental Health and Neuroscience, Maastricht University, 6229ER Maastricht, The Netherlands

**Keywords:** traumatic spinal cord injury, cAMP, phosphodiesterases, neuroinflammation, regeneration

## Abstract

Traumatic spinal cord injury (SCI) is characterized by severe neuroinflammation and hampered neuroregeneration, which often leads to permanent neurological deficits. Current therapies include decompression surgery, rehabilitation, and in some instances, the use of corticosteroids. However, the golden standard of corticosteroids still achieves minimal improvements in functional outcomes. Therefore, new strategies tackling the initial inflammatory reactions and stimulating endogenous repair in later stages are crucial to achieving functional repair in SCI patients. Cyclic adenosine monophosphate (cAMP) is an important second messenger in the central nervous system (CNS) that modulates these processes. A sustained drop in cAMP levels is observed during SCI, and elevating cAMP is associated with improved functional outcomes in experimental models. cAMP is regulated in a spatiotemporal manner by its hydrolyzing enzyme phosphodiesterase (PDE). Growing evidence suggests that inhibition of cAMP-specific PDEs (PDE4, PDE7, and PDE8) is an important strategy to orchestrate neuroinflammation and regeneration in the CNS. Therefore, this review focuses on the current evidence related to the immunomodulatory and neuroregenerative role of cAMP-specific PDE inhibition in the SCI pathophysiology.

## 1. Introduction

Traumatic spinal cord injury (SCI) is a neurological disorder characterized by damage to the spinal cord leading to transient or permanent neurological deficits. The current therapies include surgical decompression, rehabilitation, and in some instances, the administration of the corticosteroid methylprednisolone [1]. Nevertheless, the use of methylprednisolone is under debate because of severe side effects such as wound infections, sepsis, and pulmonary embolism [2]. Hence, there is an urgent need for novel therapies to treat SCI, and modulating the initial neuroinflammation and boosting endogenous repair are considered crucial to improve SCI outcomes.

The SCI pathophysiology is divided into a primary and a secondary injury phase. The primary insult encompasses the initial trauma, leading to blood–spinal cord barrier (BSCB) disruption, hemorrhage, and cellular death. The secondary phase will be initiated minutes after the primary injury and is characterized by the acute phase (<2 days post injury (dpi)), subacute phase (2–14 dpi), and the chronic phase (>6 months pi) [3]. Both the acute and subacute phases are characterized by immune cell activation, leading to profound neuroinflammation, cellular death, and the spread of tissue damage, creating an environment that blocks tissue regeneration. Next, the chronic phase is distinguished by the formation of a glial scar that seals the injury and hence prevents the spread of additional damage. Although beneficial in the early phase, the glial scar will impede neuronal regeneration in the chronic phase [4]. The injury environment, along with the reduced regenerative capacity of spinal cord cells, hampers functional recovery.

Cyclic adenosine monophosphate (cAMP) is a second messenger that has been shown to control inflammatory responses and drive endogenous regeneration of neurons and oligodendrocytes [4,5]. The well-known effectors of cAMP are protein kinase A (PKA) and an exchange factor directly activated by cAMP (EPAC) [6]. In the context of neuroinflammation, PKA activates the cAMP-response element binding protein (CREB) by phosphorylation, after which CREB activates the transcription of genes involved in anti-inflammatory phagocyte polarization (arginase-1 (Arg1), interleukin-10 (IL-10)) and genes implicated in stimulating neuroregeneration such as brain-derived neurotrophic factor (BDNF) [7,8,9]. EPAC-mediated signaling activates the downstream molecule mitogen-activated protein kinase (MAPK) and extracellular signal-regulated kinase (ERK) 1/2 pathway, which suppresses inflammation and is entangled in neuronal survival [10,11]. Furthermore, PKA and EPAC signaling inhibit translocation of the inflammation-stimulating transcription factor nuclear factor kappa B (NF-κB) and hamper growth cone collapse and neuronal apoptosis [12,13,14]. Therefore, cAMP signaling is multifunctional and involved in different processes that influence the SCI pathophysiology. Hence, fine-tuned modulation of cAMP levels is a promising strategy to tackle neuroinflammatory and neuroregenerative processes in SCI.

Nine transmembrane adenylate cyclase (AC) subtypes are responsible for cAMP production from adenosine triphosphate, which are regulated by a diversity of mechanisms [15,16]. G protein-coupled receptors (GPCRs) are the predominant regulators of AC-mediated cAMP regulation. GPCRs have α, β, and γ subunits with a broad heterogeneity in their subunits. Consequently, a compelling variety in cellular signaling through GPCRs is achieved by combining different subunits [17]. GPCR-dependent cAMP signaling is involved in the inflammatory response. For instance, chemokines are released shortly after the initial spinal cord insult and attract inflammatory cells such as phagocytes and lymphocytes to the spinal cord lesion. Multiple chemokines, e.g., chemokine C-X3-C motif ligand 1 (CX3CL), chemokine C-C motif ligand 2 (CCL2), or CCL3, direct immune cell migration to the place of CNS trauma through binding to their GPCRs. Receptor signaling of chemokine receptors predominantly inhibits AC activity through the Gα_i_ subunit and hence reduces cAMP levels [18,19]. Furthermore, GPCR signaling regulates immune cell activation and thus controls the production of cytokines. For example, complement system mediators influence immune cell phagocytosis and inflammatory cytokine production through GPCRs. Here, complement 3a receptor (C3aR) and C5aR lower cAMP levels through Gα_i_-mediated AC inhibition, resulting in NF-κB translocation and the induction of pro-inflammatory cytokine expression [20]. Additionally, physiological elevations of intracellular Ca^2+^ levels inhibit the AC activity of subtypes AC5 and AC6 through calmodulin in the central nervous system (CNS). However, a non-physiological elevation of Ca^2+^, as in SCI, inhibits all transmembrane ACs [16,21]. Moreover, Ca^2+^ regulates soluble AC10, but its role remains largely unknown in the CNS [22]. Lastly, endogenous proteins, DNA, and modified lipids released from the damaged spinal cord are recognized by toll-like receptors (TLRs), leading to immune cell activation. TLR (e.g., TLR4) activation stimulates the production of protein kinase C (PKC), which can modulate cAMP levels via AC inhibition [23,24]. The binding of inflammatory cytokines tumor necrotic factor alpha (TNF-α) and IL-1β to their receptors mediates cAMP responses through activation of PKC, resulting in AC2 and AC5 inhibition [25,26].

Besides immune cell modulation, promoting neuroregeneration is crucial to improve functional outcomes following SCI. Neuroregeneration can be stimulated by either reviving endogenous repair of neurons or promoting the remyelination capacity of oligodendrocytes or infiltrated Schwann cells (SCs). The majority of the neuroregenerative properties of neurotrophins such as BDNF, neurotrophin-4 (NT-4), glial-derived neurotrophic factor (GDNF), or neurotrophic growth factor (NF) are regulated by binding to the tropomyosin receptor kinase (Trk) receptor family or the complex formed between Trks and p75 neurotrophin receptor (p75^NTR^) [27]. The binding of neurotrophins induces the activation of the EPAC downstream signaling molecule ERK1/2, leading to CREB phosphorylation and growth cone preservation in neurons [28]. Additionally, GPCRs, e.g., adenosine A_2A_ receptor or procaspase activating compound-1 (PAC1) receptor activation, modulate neuroregeneration through AC-mediated cAMP production, leading to transactivation of the Trk receptors in the absence of neurotrophins [29]. Furthermore, the Gα_s_ unit is crucial to increasing Trk receptor translocation to the cellular surface and hence enhancing Trk signaling [30]. In contrast, receptors stimulating growth cone collapse and neuronal death, e.g., Nogo receptor (NgR) in complex with p75^NTR^, induce neuronal degeneration by activating the Ras homolog gene family member A (RhoA)/Rho-associated protein kinase (ROCK) pathway [31,32]. However, RhoA is known to be inhibited by downstream molecules of PKA and EPAC. Therefore, stimulating cAMP levels can be beneficial to promoting pathways involved in neuroregeneration and inhibiting pathways for neuronal degeneration through PKA and ERK1/2.

A rapid decrease in cAMP levels is evident in SCI lesions and their surrounding tissues [33,34,35]. As mentioned above, AC is an important regulator for cAMP production, but its widespread expression and the limited number of subtypes impede ACs as a target to regulate cAMP in a cell-specific and controlled manner. Spatiotemporal control of cAMP is achieved by phosphodiesterases (PDEs). PDEs are a class of enzymes responsible for the hydrolysis of cAMP and cyclic guanosine monophosphate (cGMP). PDEs are clustered into 11 gene families (PDE1-11) and have substrate specificity for either cGMP (PDE5, 6, and 9), cAMP (PDE4, 7, and 8), or dual specificity (PDE1, 2, 3, 10, and 11). Each PDE gene encodes for multiple subfamilies (e.g., PDE4A-D) and even more isoforms (e.g., PDE4B1-5) with a total of 77 different isoforms, yielding a cell-specific PDE isoform fingerprint [36,37]. The PDE4 family has a high expression in the murine and human CNS, with the exception of the PDE4C subfamily. A comprehensive overview of PDE expression in the CNS is provided in the review of Rombaut et al. [38]. In short, PDE4A shows a low expression in the human CNS but is abundantly present in murine astrocytes compared to the other subfamilies. In addition, PDE4B has a high expression in murine astrocytes, murine oligodendrocytes, and human microglia. Lastly, PDE4D is highly expressed in human and murine astrocytes and neurons and mouse oligodendrocyte precursor cells (OPCs). PDE7 and PDE8 are also expressed in the CNS. PDE7A is the predominant subfamily expressed in oligodendrocytes, astrocytes, and neurons of mice and human astrocytes, while PDE7B is mainly expressed in murine astrocytes and human astrocytes and neurons. Lastly, PDE8A and PDE8B are expressed in murine oligodendrocytes, while only PDE8B is expressed in higher levels in murine and human neurons [38,39]. The spatiotemporal control of PDE expression provides the opportunity to target specific cAMP-mediated cellular processes in CNS injury, including SCI. Hence, the aim of this review is to discuss the effect of cAMP substrate-specific PDE inhibition on the neuroinflammatory and neuroregenerative processes following rodent SCI.

## 2. cAMP-Specific PDEs in Neuroinflammation following SCI

The neuroinflammatory environment in the spinal cord lesion is initiated through the activation of resident phagocytes and infiltrated immune cells such as neutrophils, monocytes, and lymphocytes. Diminishing spinal cord neuroinflammation by either reducing cellular activation, hampering immune cell infiltration, or skewing immune cell phenotype is key to create a regeneration-permissive environment and hence ameliorate functional recovery after SCI. cAMP modulation can be an excellent candidate to orchestrate neuroinflammation through PKA/EPAC-dependent mechanisms. Therefore, the following sections will outline the role of cAMP-specific PDEs in the different immune cells involved in neuroinflammation after SCI.

### 2.1. Neutrophils

Neutrophils are the first circulating cells of the innate immune system to infiltrate the injured spinal cord and peak in number 24 h pi [40]. Shortly after the arrival in the spinal cord, neutrophils start releasing pro-inflammatory products such as myeloperoxidase, elastases, and matrix metalloproteinase (MMP) 9, which contribute to the spread of tissue injury. Additionally, neutrophils form neutrophil extracellular traps (NETs) composed of neutrophil DNA and granular proteins. These NETs promote neuroinflammation and neuronal apoptosis in the SCI lesion of rats [41]. Accordingly, lowering neutrophil infiltration in the spinal cord lesion improved locomotor outcomes in murine contusion SCI [42]. Nevertheless, the complete depletion of neutrophils worsens neurological outcomes of spinal cord-injured mice [43]. The potential beneficial role of neutrophils in SCI is attributed to the neutrophil-mediated phagocytosis of compounds that inhibit regeneration, such as cellular and myelin debris in the SCI lesion. Hence, removing these anti-regenerative compounds promotes the formation of a pro-regenerative environment for neurons and oligodendrocytes [44,45]. However, the beneficial role of neutrophils remains to be elucidated in SCI, but it is suggested that modulating neutrophils is an important target for defining SCI outcomes.

cAMP is a pivotal regulator of neutrophil activity, e.g., phagocytosis, NET formation, and the release of inflammatory products [46,47,48]. Orchestrating cAMP levels upon a bolus intravenous (i.v.) injection of pan PDE4 inhibitor IC486051 2–60 h after a compression SCI reduced neutrophil infiltration and MPO activity 24 and 72 h pi in rats. Interestingly, these effects were only observed in rats treated with 0.5 mg/kg or 1 mg/kg IC486051, while only limited effects were observed with a high dose of 3 mg/kg [49]. The limited anti-inflammatory effect of 3 mg/kg IC486051 can be due to off-target effects observed at a high dose of PDE4 inhibition [50,51]. PDE4B was revealed to be an especially crucial mediator for neutrophil infiltration in the CNS, since intraperitoneal (i.p.) administration of 3 mg/kg A33, 30 min and 5 h pi reduced neutrophil infiltration in traumatic brain injury [52]. Nevertheless, in this specific study, A33 treatment was not compared to pan PDE4 inhibition, and hence, the contribution of other PDE4 isoforms in regulating neutrophil infiltration cannot be excluded. Indeed, the administration of 3 mg/kg pan PDE4 inhibitor rolipram could further decrease neutrophil infiltration in the lungs of lipopolysaccharide (LPS)-induced chronic obstructive pulmonary disease (COPD) in PDE4B knock-out (KO) mice and, to a lesser extent, in PDE4D KO mice, suggesting a promising role of PDE4D in orchestrating neutrophil infiltration [53]. However, the role of individual PDE4 families in regulating neutrophil infiltration over the BSCB should be further investigated since the mechanisms of neutrophil infiltration depend on the surrounding tissue barrier. Additionally, PDE4 inhibition can alter the activity of peripheral neutrophils, leading to modified infiltration rather than PDE4 inhibition directly modulating neutrophil–BSCB interaction [54]. In vitro, pan-PDE4 inhibition with 0.01–1000 nM roflumilast suppressed the production of tissue-damaging compounds MPO, elastase, and MMP-9 in human neutrophils [55]. The ability of cAMP to cease neutrophil activation is well known and is suggested to be mediated through PKA [56]. In conclusion, PDE4 inhibition has the ability to cease neutrophil infiltration and impede the neutrophil-mediated spread of tissue damage by suppressing neutrophil activity.

PDE7 is also expressed in human neutrophils and could, therefore, be involved in regulating specific functions [57]. Research showed that PDE7 inhibition 1, 3, or 6 h pi with 4 mg/kg VP1.15 (i.p.) or 10 mg/kg S14 (i.p.) reduced neutrophil infiltration 24 h pi in mice with compression-induced SCI [58]. Nevertheless, neutrophil infiltration was measured by quantifying MPO activity, and hence, the decrease in MPO activity could be attributed to diminished neutrophil activation rather than reduced infiltration. Indeed, PDE7 was indicated to act in an immunomodulatory way by decreasing TNF-α, IL-1β, inducible nitric oxide synthase (iNOS), and cyclooxygenase 2 (COX-2) levels in the spinal cord tissue 24 h pi as quantified with immunohistochemistry. However, the pro-inflammatory products in the spinal cord lesion did not colocalize with neutrophils. Therefore, the effect of PDE7 inhibition on pro-inflammatory cytokine production can potentially be attributed to influencing other immune cells, such as infiltrated monocytes, microglia, or lymphocytes in the spinal cord lesion [58]. Consequently, PDE7 inhibition is suggested to modulate neuroinflammation in SCI, but the neutrophil-specific influence of PDE7 inhibition in the spinal cord lesion remains largely unknown.

### 2.2. Phagocytes

The secondary injury in SCI is characterized by the profound activation of CNS phagocytes. Microglia are the resident immune cells of the CNS and rapidly migrate to the spinal cord lesion, adopting a phenotype recognized by the production of TNF-α and IL-1β [59]. Circulating phagocytes, namely, monocyte-derived macrophages (MDMs), are also attracted to the spinal cord lesion. MDMs will contribute to the production of inflammatory cytokines, e.g., TNF-α and IL-1β, along with the resident microglia [60,61,62]. Together, the phagocytes are major mediators of tissue damage by creating an inflammatory environment via the secretion of cytokines and reactive oxygen species. Additionally, infiltrated MDMs contact injured axons and mediate axonal dieback [63,64]. On the other hand, phagocytes can also promote tissue regeneration by clearing anti-regenerative compounds such as cellular debris and producing neurotrophic factors, e.g., BDNF, NGF, and NT-3, to stimulate neuroregeneration [65,66]. Microglia are the primary phagocytes removing tissue debris in the lesion site in the early stages of SCI, whereas MDMs are the predominant phagocytes in the later stages of SCI [60]. However, MDMs are less capable of processing myelin debris, leading to the formation of inflammatory foamy macrophages [67]. Both repair-promoting and inflammatory phagocytes are observed shortly after the initial injury. The repair-promoting population declines after the first week following the initial injury, while the inflammation-stimulating phagocytes reside until the chronic phases of SCI [60,61]. The persistent inflammatory phagocytes and the decline in regeneration-promoting phagocytes contribute to profound neuroinflammation followed by tissue damage in the spinal cord. Hence, endogenous regeneration in the SCI tissue is diminished, leading to impaired recovery following SCI.

Reducing MDM infiltration and ceasing the production of inflammatory compounds of spinal cord phagocytes has been shown to be promising in promoting functional recovery following SCI [68,69,70]. Monocyte infiltration, quantified by the number of CD68^+^ cells, was significantly decreased 2 weeks pi upon i.v. administration of 1 mg/kg PDE4 inhibitor rolipram starting 1, 4, or 24 h pi and thereafter once daily for 2 weeks in a thoracic SCI in rats [71]. Elevating cAMP levels stimulated anti-inflammatory cytokine production (Arg1, chitinase-3-like protein 3 (Ym-1), and IL-10) of phagocytes and reduced LPS-induced TNF-α and IL-1β release of the murine microglia BV2 cell line [72,73,74,75]. The anti-inflammatory effect of 10 µM PDE4 inhibitor FCPR03 on BV2 cells was suggested by the activation of the PKA/CREB pathway since PKA inhibition counteracted PDE4-induced inhibition of TNF-α and IL-1β production. Additionally, PDE4 inhibition with 10 µM of rolipram in the EOC-2 murine microglia cell line or intragastrical treatment with 0.5–1 mg/kg FCPR03 in vivo lowers NF-κB-p65 phosphorylation and NF-κB activity in the hippocampus of LPS-injected mice, respectively [76,77]. Consequently, PDE4 inhibition is considered to reduce inflammatory responses of phagocytes by promoting anti-inflammatory cytokine production and hampering pro-inflammatory cytokine production through the PKA/CREB pathway and PKA-induced inhibition of NF-κB, which can be beneficial to modulating phagocyte responses in SCI. The immunomodulatory effects of PDE4 inhibition on phagocytes were further observed in SCI animal models. Subcutaneous (s.c.) delivery of 0.5 mg/kg/day pan PDE4 inhibitor rolipram for 1 week reduced the production of pro-inflammatory cytokines TNF-α, 3 and 6 h pi in a contusion rat SCI model [33]. In contrast, 0.5 mg/kg rolipram treatment could not reduce IL-1β levels, suggesting differential regulation in cytokine production in vivo. Additionally, research showed that PDE4 inhibition by i.p. administration of 0.5–1 mg/kg roflumilast 30 min before SCI induction increased the amount of CD163^+^ microglia and decreased the amount of CD68^+^ microglia 28 pi. The increase in repair-promoting phagocytes (CD163^+^ phagocytes) was accompanied by a reduction in pro-inflammatory cytokines TNF-α and IL-1β and enhanced anti-inflammatory cytokine (IL-10) production in rats with a compression-induced SCI [78].

In particular, PDE4B is of interest to modulate phagocyte activity and circumvent emetic side effects. PDE4B deficiency in mouse peritoneal macrophages enhanced the secretion of anti-inflammatory interleukin-1 receptor antagonist (IL-1Ra) through a cAMP/PKA but not in an EPAC-dependent manner [79]. PDE4B is acutely upregulated in both microglia and macrophages following SCI in mice with, in particular, high PDE4B2 mRNA expression [34,80]. Moreover, the rise in PDE4B coincided with elevated mRNA expression of TNF-α and IL-1β, which supports PDE4B-mediated regulation of pro-inflammatory cytokine production in SCI [80]. Although the PDE4B inhibitory effect on SCI is unknown, recent data of murine autoimmune encephalomyelitis (EAE) revealed that twice daily s.c. treatment with 3 mg/kg PDE4B inhibition shifts the balance to Arg-1^+^ macrophages but could not reduce monocyte infiltration in the spinal cord (for a comprehensive overview of PDEs in MS, see the review of Schepers et al.) [81,82]. Therefore, phagocyte infiltration in the spinal cord is potentially attributed to other PDE subtypes or the combination of different PDE4 genes, since general PDE4 inhibition could reduce monocyte infiltration in the spinal cord after SCI [83].

PDE7 is expressed to a lesser extent in microglia and macrophages [38]. Nevertheless, i.p. administration of 4 mg/kg VP1.15 or 10 mg/kg S14 1, 3, and 6 h pi and daily for 9 days reduced pro-inflammatory cytokine production potentially by a decreased degradation of IkappaB kinase (IκB-a), ERK, p38, and Jun N-terminal kinase (JNK) in a compression SCI mouse model [58]. Although this was not directly measured in spinal cord phagocytes, the MAPK and ERK1/2 pathway is postulated as key regulator of phagocyte cytokine production [84]. Furthermore, PDE7 inhibition with 10 mg/kg VP3.15 or 20 nmol S14 reduced microgliosis in a murine EAE model or LPS-induced Parkinson’s disease model in rats [85,86]. Therefore, PDE7 can be a key target to modulate cAMP levels in SCI to alter phagocyte activation.

### 2.3. Lymphocytes

T and B lymphocytes will infiltrate the spinal cord in the first week following SCI [87]. The lymphocyte phenotype in the SCI lesion depends on the surrounding environment and can either promote the inflammatory processes or stimulate regeneration [40]. The damaged SCI environment attracts CD4^+^ T lymphocytes to produce pro-inflammatory cytokines, leading to phagocyte activation and profound tissue damage. Furthermore, it is well known that lymphocytes demonstrate autoimmunity against CNS components such as myelin basic protein (MBP) following SCI [88]. Although the number of lymphocytes will decrease over time, they remain high in number but lower than the spinal cord macrophages over 14 days of injury [87,89]. Autoreactive B cells produce auto-antibodies and pro-inflammatory cytokines, stimulating the inflammatory and tissue-damaging cascade following SCI. The importance of both T and B cells is stressed when specific T- and B-cell subsets are depleted in SCI. Depletion of Vγ4 γδ T cells, which are responsible for phagocyte stimulation via IFN-γ production, improved functional recovery and lowered immune responses following SCI [90]. Additionally, B-cell-depleted mice or B-cell depletion with CD20 antibodies improved locomotor outcomes in mice, reduced neuroinflammation, and acted neuroprotective in an in vivo murine SCI model [91,92]. Therefore, modulating lymphocyte activation and reducing autoreactivity against spinal cord compounds is considered a promising strategy to influence the neuroinflammatory processes after SCI.

Antigen-induced activation of both B and T lymphocytes is modulated by cAMP levels [93]. Additionally, cAMP orchestrates antigen-stimulated proliferation and antibody production in B cells as well as the maintenance of naive T cells and their activation [94]. Furthermore, T-cell activation is hampered upon increasing cAMP levels, and its elevation is crucial in the formation of regulatory T cells. However, prolonged high cAMP levels induce an anergy-like state [95,96,97]. PDE-mediated modulation of lymphocytes in SCI has not been investigated yet, but in vitro works suggest the promising implications of PDE inhibitors to lower lymphocyte activation in SCI. One micromole of PDE4 inhibitor roflumilast and its metabolite roflumilast N-oxide reduced the proliferation of spleen-derived murine CD4^+^ T cells stimulated with anti-CD3/CD28 antibody. Roflumilast was suggested to block the interaction between inositol triphosphate 3 (IP3) and its receptor, leading to decreased calcium release from the Golgi apparatus and resulting in diminished activation of the nuclear factor of activated T cells (NFAT) and ceased IL-2 transcription, necessary for T-cell proliferation [98]. Furthermore, 1 µM of PDE4 inhibitor RP73401 decreased T-cell proliferation and skewed them to an anti-inflammatory phenotype. Selective PDE4D KO with siRNA showed that the PDE4D subtype is particularly crucial for regulating T-cell proliferation, while PDE4B and PDE4A were less important. However, PDE4A, PDE4B, and PDE4D were all demonstrated to modulate IL-2, IFN-γ, and IL-5 levels in human primary CD4^+^ T cells [99]. PDE4B2 is suggested to mediate T-cell receptor-induced IL-2 production in Jurkat T cells [100]. IL-2 is a cytokine involved in the proliferation and differentiation of effector T cells and regulatory T cells. Hence, it remains to be elucidated whether the production of IL-2 in response to PDE4B2 upregulation is either detrimental by stimulating the expansion of effector T cells or beneficial to prevent autoimmunity with the production of regulatory T cells. PDE4B upregulation during T-cell receptor (TCR) stimulation in human-isolated CD4^+^ T lymphocytes has been demonstrated to be the main contributor to reducing cAMP levels. Interestingly, PDE4B mRNA levels decreased after 24 h while PDE4A and PDE4D mRNA levels were upregulated and peaked 120 h after anti-CD3/CD28 stimulation in CD4^+^ T cells, indicating a time-dependent response [99]. To conclude, different PDE subfamilies are involved in variable processes in the adaptive immune system since PDE4D was shown to be involved in T-cell proliferation while PDE4B was demonstrated to be a major regulator of T-cell activation.

Little is known about the effects of PDE7 and PDE8 inhibition on lymphocytes in SCI. However, PDE7 inhibition was revealed to be beneficial for reducing T-cell proliferation, ceasing IL-17 production, and expanding Foxp3 (T-regulatory) expression in an EAE model of MS [85,101,102]. PDE8 inhibition with 10 mg/kg s.c. PF-04957325 thrice daily for 6–13 days reduced immune cell infiltration of CD4^+^ T cells 13 days after EAE induction with myelin oligodendrocyte glycoprotein (MOG) [103]. Moreover, PDE8 inhibition with s.c. 2.5 mg/kg of PF-04957325 could not reduce T-cell proliferation in an allergic airway disease mouse model but could regulate T-cell effector interactions with the endothelial cells (for a comprehensive review of the effect of PDE7 and PDE8 inhibition on lymphocytes in MS, please read the review of Schepers et al.) [81,104].

## 3. cAMP-Specific PDEs in Neuroregeneration following SCI

The initial spinal cord insult elicits the cellular death of neurons, oligodendrocyte precursor cells (OPCs), and oligodendrocytes. Subsequently, endogenous regeneration is hampered by the inflammatory CNS environment. Consequently, new therapies should focus on reactivating the regenerative processes of spinal cord cells to improve functional recovery in SCI patients. Therefore, the following sections will describe the potential of cAMP-specific PDEs in promoting neuroregeneration following SCI.

### 3.1. Neurons

The initial injury and activation of the inflammatory responses following SCI lead to the death of spinal cord motor neurons. Additionally, axonal dieback is mediated by infiltrated macrophages, leading to axonal retraction (reviewed by Van Broeckhoven et al.) [68]. Important factors in the spinal lesion that inhibit neuronal and axonal regeneration are proteins released from injured oligodendrocytes, including myelin-associated glycoprotein (MAG) and oligodendrocyte myelin glycoprotein (OMgp). In addition, Nogo-A is present on axons as well as released from neurons and is considered a major inhibitor for neurite outgrowth [31]. Together, myelin-associated proteins and neuronal inhibitory molecules induce neuronal apoptosis and growth cone collapse by activating the RhoA/ROCK pathway [31,32]. Elevating cAMP levels is promising to impede neuronal apoptosis and promote neuronal regeneration through PKA/EPAC-mediated inhibition of RhoA, which can be indispensable for improving functional recovery for SCI patients.

While MAG inhibits neuronal regeneration in adult neurons, MAG does not inhibit axonal outgrowth during certain embryonal stages. Cai et al. demonstrated that MAG-induced inhibition of axonal regeneration was correlated with cAMP levels in neurons [105]. This concept was reinforced since elevating cAMP in adult neurons could overcome the MAG-induced inhibition of neurite outgrowth suggested by cAMP-mediated activation of the PKA pathway [106,107]. Furthermore, glial-released neurotrophins promote neuronal outgrowth by elevating cAMP-directed activation of ERK signaling [14,107]. More specifically, ERK-induced inhibition of PDE4 led to elevated cAMP levels and proved to be crucial to overcoming MAG-mediated inhibition upon the presence of BDNF [108]. Consequently, modulating PDE activity in SCI could be useful to promote neuronal regeneration and hence improve functional recovery in SCI patients. Treatment with 1 µM roflumilast or s.c. administration of 3 mg/kg roflumilast increased neuronal viability in vitro and in a stroke model, respectively [109,110]. The decrease in neuronal cell death by roflumilast is postulated through decreasing pathways associated with neuronal cell death upon endoplasmic reticulum (ER) stress, such as the inositol-requiring enzyme 1 a (IRE1a)/JNK pathway [111]. Another protective mechanism of roflumilast in neurons could be through inhibition of NF-κB, since 21 days of treatment with 2 mg/kg orally administered roflumilast was neuroprotective against quinolinic acid administration in mice through inhibiting NF-κB and elevating CREB and BDNF expression in the striatum and cortex in rats [112]. Inhibition of PDE4B with i.p. injections of 0.3 mg/kg A33 decreased neuronal loss in traumatic brain injury [52]. Nevertheless, PDE4B is shown to be upregulated in inflammatory cells following traumatic CNS injury, and therefore, the neuroprotective effects of PDE4B inhibition could be achieved indirectly through decreased neuroinflammation. The neuroprotective role of PDE4 inhibition was further confirmed in SCI. First of all, i.v. administration of 1 mg/kg rolipram starting 2 h pi thereafter, continued daily for 2 weeks, promoted neuronal and axonal protection at the end of treatment in a rat thoracic contusion model of SCI [71]. Similar neuroprotective effects were revealed 4 weeks pi upon 0.4 μmol/kg/h s.c. administration of rolipram for 10 days starting 14 dpi in a cervical hemisection model with embryonic tissue transplantation in the lesion site. An embryonic transplant was used to assess axonal growth into the transplant since there is usually no ingrowth in the transplant, but ingrowth was observed during rolipram inhibition [113]. Furthermore, PDE4 inhibition promoted neuronal regeneration in the inhibitory environment of SCI. Here, 0.1, 0.25, 1, or 2 µM of rolipram treatment could overcome MAG-inhibited neurite outgrowth in isolated dorsal root ganglia (DRG) neurons of rats [113]. In addition to the inflammatory environment, chondroitin sulfate proteoglycans (CSPs), produced by glial cells, are pivotal in inhibiting neuronal regeneration in the subacute and chronic phases of SCI. Rolipram alone was unable to promote neuronal survival and outgrowth in a CSP-rich environment in vitro. In addition, it was not potent in protecting neurons against glutamate-induced neuronal toxicity [114]. Hence, PDE4 inhibition can potentially overcome the inhibitory influences of myelin and neuronal released signals. However, current evidence suggests it cannot overcome glutamate and CSP-induced neuronal outgrowth inhibition.

Little is known regarding other cAMP-specific PDEs, including PDE7 and PDE8, in neuronal regeneration. However, research suggests that PDE7 can be promising in stimulating neuronal regeneration. Pretreatment with PDE7 inhibitor TC3.6 was demonstrated to improve the viability of the neuronal cell line PC12 upon glutamate exposure [102]. In addition, the PDE7 inhibitors S14 or BRL50481 reduced 6-hydroxydopamine (OHDA)-induced cellular death of a dopaminergic cell line SH-Sy5Sy 16 h after incubation. The aforementioned PDE7 inhibitors were able to protect neurons in vivo after LPS injection in the substantia nigra [86].

### 3.2. Myelinating Cells

Oligodendrocytes are the myelinating cells in the CNS and support the spinal cord axons to facilitate saltatory transduction of signals between different body regions and the brain. Neuronal function is severely disrupted during SCI because of oligodendrocyte death by the initial injury. Subsequently, oligodendrocyte apoptosis and necrosis are continued in later phases of SCI because of the cytotoxic compounds in the secondary injury phase and loss of trophic factors by degenerating neurons [115]. Mature oligodendrocytes can be replaced in the spinal cord by oligodendrocyte precursor cells (OPCs), which start proliferating shortly after the initial injury and peak in proliferative capacity 2 weeks pi [116]. However, the loss of oligodendrocytes releases inhibitory substances, e.g., MAG and OMgp, which inhibit OPC maturation and hence remyelination [117,118]. The peripheral myelinating cells, namely, SCs, have been shown to infiltrate the spinal cord lesion. SCs have broad functions in the peripheral nervous system (PNS), including the ensheathment of neurons and the production of neurotrophins and extracellular matrix (ECM) components. These characteristics contribute to the enhanced regenerative capacity of the PNS compared with the CNS. The regenerative stimulating capacity of infiltrated SCs in the spinal cord lesion were shown to be neuroprotective and contribute to remyelinating damaged axons after SCI. Nevertheless, SCs transplantation in the spinal cord lesion appeared to be insufficient to promote functional repair in clinical trials with SCI patients [119,120,121]. Therefore, current research focuses on boosting the regenerative stimulating capacity of SCs in SCI by combining SC transplantation with different modulates, e.g., scaffolds or growth factors. Recently, cAMP modulation by PDE inhibition has been demonstrated to promote the regenerative features of SCs in vitro and improve SCI repair after SC transplantation [33]. As a consequence, orchestrating cAMP levels via PDE inhibition can be considered a promising strategy to impede functional loss following SCI via modulating oligodendrocytes and SCs.

cAMP levels regulate OPC differentiation toward the mature myelinating oligodendrocytes [118]. Although OPC migration and proliferation are observed shortly after the SCI, maturation and differentiation are hindered by myelin products released from dying oligodendrocytes. Oligodendrocyte-derived compounds such as MAG reduce phosphorylation and activation of the ERK1/2/p38/MAPK and CREB pathways in OPCs. Elevating cAMP with 3.21 mM dibutyryl (db)-cAMP or inhibiting PDE4 with 0.5 µM rolipram could oppose this myelin-induced OPC inhibition 48 h after administration in primary rat OPCs. Additionally, OPC differentiation and axonal remyelination were stimulated in focal-induced demyelination upon treatment with 0.5 mg/kg rolipram 14 dpi [118]. Besides the vital role of PDE4 inhibition in OPC differentiation, it has been demonstrated to promote oligodendrocyte protection in the SCI lesion. Whitaker et al. indicated increased oligodendrocyte sparing 24 and 72 h pi in contusive SCI in rats after administering 0.5 mg/kg/day rolipram using a mini-osmotic pump [122]. Additionally, 14 days of s.c. PDE4 inhibition with 0.5 mg/kg/day rolipram improved oligodendrocyte sparing in the white matter of the ventrolateral funiculus in a contusive rat SCI model 5 weeks pi [123]. This was further supported by Schaal et al., who demonstrated oligodendrocyte and white matter sparing upon 2 weeks of i.v. 1 mg/kg rolipram treatment in SCI [71]. PDE4 inhibition is indicated to influence SC functioning and can therefore have the potential to boost SC function in the spinal cord. Six days of treatment with 10 µM roflumilast on rat-derived SCs, followed by co-culturing human iPSC-derived nociceptive neurons demonstrated an increased myelinated area and enhanced neurite outgrowth after 14 days through EPAC and PKA-mediated mechanisms, respectively. Interestingly, the effects of PDE4 inhibition on myelination and axonal outgrowth were not observed after 21 days, indicating that PDE4 inhibition can accelerate the remyelinating and mitogenic activity of SCs [124]. The beneficial effects of boosting SC functioning with PDE4 inhibition were observed in SCI. S.c. administration of 0.5 mg/kg/day rolipram starting immediately after a T8 contusion injury in rats, followed by a SC transplantation 1 week pi improved axonal sparing and the number of SC-myelinated axons in the spinal cord lesion 11 weeks pi. Interestingly, rolipram treatment without SC transplantation was shown to increase peripheral myelin sheaths, suggesting a myelinating boosting effect of PDE4 inhibition on SC. Acute rolipram treatment combined with SC transplantation was not able to improve serotonin^+^ (5HT) fiber ingrowth in the SC transplant graft. In contrast, the combination of acute rolipram treatment with SC transplantation and db-cAMP administration 1 week pi improved 5HT ingrowth in the graft. It was postulated that the acute and local elevation of cAMP was responsible for the production of neurotrophic factors and modulation of the CNS environment that could not be achieved by the daily PDE4 inhibition [33].

To circumvent the emetic side effects, subtype-specific PDE4 inhibition has shown already promising results. It was shown that PDE4B KO mice had increased white matter sparing and higher levels of oligodendrocytes 42 dpi in a contusion model T9 [80]. However, PDE4B is mainly involved in regulating inflammatory processes. Consequently, the increase in oligodendrocyte sparing can potentially be attributed to decreased neuroinflammation and hence an improved oligodendrocyte sparing and recovery. PDE4D is highly expressed in oligodendrocytes, and pan PDE4 with 1 µM of roflumilast and PDE4D inhibition with 0.5 or 1 µM of GEBR32a, but not PDE4B inhibition with 1 µM of A33, was shown to increase OPC maturation in vitro [82].

PDE7 inhibition with 10 mg/kg VP3.15 could also increase OPC differentiation after 15 days of treatment in Theiler’s murine encephalitis virus-induced demyelinating disease [85]. However, the numbers of OPCs were not increased, indicating that PDE7 inhibition potentially does not induce OPC proliferation but rather stimulates oligodendrocyte recovery. Additionally, PDE7 inhibition in an EAE mouse model of MS showed decreased demyelination, indicating a neuroprotective effect, either direct or indirect by immune modulation, of PDE7 inhibition [102]. These results suggest the promising role of PDE7 inhibition in oligodendrocyte survival, therefore providing prolonged survival of oligodendrocytes in SCI, but this remains to be elucidated.

### 3.3. Astrocytes

Astrocytes are the most abundant glial cells in the CNS and are important regulators of CNS water content, glutamate concentration, and maintenance of ionic balances [125]. Following SCI, astrocytes become activated by the disrupted spinal cord environment and adopt a ramified morphology. Subsequently, activated astrocytes start proliferating and surround the lesion site to form a glial scar that seals the injury. Although the glial scar is initially neuroprotective by reducing the spread of tissue damage, it is also considered a major barrier for neuronal regeneration. Activated astrocytes contribute to the production of inflammatory cytokines and chemokines to promote immune cell infiltration and activation [126]. Therefore, astrocytes play a crucial role in the initial inflammatory reaction in SCI but are also considered protective owing to their contribution to the glial scar.

Cytokines released from astrocytes can skew phagocyte polarization and hence influence the inflammatory environment [40]. cAMP levels have proved to be an important regulator of the homeostatic functions of astrocytes since an elevation in cAMP causes an upregulation of genes involved in homeostatic control, metabolic function, and antioxidant mechanisms [125]. Embryonic implanted tissue in the SCI lesions of rats was used to assess the role of PDE4 inhibition on astrocyte activation in the lesion site. S.c. administration of 0.4 µmol/kg rolipram in SCI was demonstrated to reduce astrogliosis in embryonic implanted tissue in the SCI lesion site and surrounding spinal cord tissue, indicating reduced astrocyte activity after SCI upon PDE4 inhibition [113].

PDE7 inhibition reduced astrocyte nitrite, TNF-α, and COX-2 production following LPS stimulation in vitro, and this downregulation of pro-inflammatory cytokines was mediated by the PKA pathway [127]. The astrocyte inhibitory effect of PDE7 in vitro reveals a promising implication of PDE7 inhibition to impede astrocyte activation in SCI. However, in vivo research is demanded to reveal the role of PDE7 inhibition in SCI.

In addition to the inflammatory contribution of astrocytes in SCI, astrocytes exhibit protective properties following SCI by sealing the injury starting 1 dpi to prevent the spread of tissue damage [126,128]. This protective role of astrocytes is further demonstrated since complete ablation results in increased immune cell infiltration and activation, neuronal degeneration, and demyelination, leading to reduced functional recovery in rodents [129]. Although protective, astrocytes adopt a scar-forming phenotype and produce chondroitin sulfate proteoglycans, which are the main inhibitors of OPC growth and differentiation, neuronal outgrowth, and in general, regeneration [126,128,130]. The produced CSPs trigger growth cone collapse mediated through the RhoA/ROCK pathway [131]. Elevating cAMP levels can potentially reduce the glial scar in later phases of SCI since their elevation in astrocytes reduces gene expression for the production of glial scar products, including CSPs [132]. Although little is known about the interaction of PDE inhibition and CSPG, the elevation of cAMP’s downstream molecule EPAC increases neurite outgrowth in postnatal rat cortical neurons in vitro. EPAC agonists were shown to reduce astrocyte activation in vitro during LPS incubation and elongated astrocytes at the lesion border, which guided axonal outgrowth. EPAC agonists overcame CSPG inhibition of neuronal growth in an ex vivo model of SCI [133]. Nevertheless, current evidence indicated that PDE4 inhibition alone was not able to lower CSP production or induce CSP breakdown in an in vivo model of contusive SCI in rats [114]. Both neuroinflammation and neuroregeneration in the spinal cord are predominantly visualized using immunofluorescence. Therefore, an overview of commonly used fluorescent markers for these processes is described in Table 1.

## 4. Functional Recovery

cAMP-specific PDE inhibition has been demonstrated to influence neuroinflammation and regeneration and could, therefore, be a promising intervention to promote SCI outcomes. Indeed, s.c. delivery of 3.18 mg/kg PDE4 inhibitor rolipram improved functional outcomes in rats suffering from a contusion injury. Rolipram was able to ameliorate functional outcomes 56 dpi, as shown by improved BBB scores and the ability to walk on smaller beams, which indicates improved locomotor coordination. The significant improvement in functional outcomes started 7 and 3 dpi for BBB scoring and beam walk testing, respectively [134]. Interestingly, dose-dependent effects were demonstrated following 0.4 or 0.8 µmol/kg/h rolipram treatment for 10 consecutive days starting 2 weeks pi in a cervical hemisection model of SCI. The 0.4 µmol/kg/h treatment improved functional recovery measured by rearing behavior, while this was not observed with the higher dose of 0.8 µmol/kg/h [135]. Either 0.5 mg/kg of rolipram or IC486051 improved functional recovery. First, i.v. IC486051 improved BBB scores significantly in contusion SCI in rats. Second, roflumilast reduced the percentage of hindlimb footfall errors during the grid-walking test following a compression SCI injury, indicating improved locomotor coordination. However, the timing of the administration seemed to be crucial since IC486051 injected past 60 h pi could not improve functional recovery [49,123]. In contrast, studies combining rolipram with other therapies could not find improved outcomes with rolipram alone. Here, 0.5 mg/kg/day of rolipram treatment delivered by an osmotic pump for 2 weeks starting after injury induction showed no improvement in BBB scores at the end of the study, while rolipram in combination with SC transplantation and db-cAMP enhanced functional recovery [33]. Additionally, 0.5 mg/kg/day of rolipram, delivered using a mini-osmotic pump for 14 days, could not improve functional recovery after 8 weeks pi in rats with a T9-T10 contusion injury, while rolipram administered along with a bolus of methylprednisolone was able to achieve this [114]. PDE4 inhibition is hampered for clinical use because of its severe emetic side effects. Hence, a combination of well-timed subtype-specific inhibition can be a promising strategy in modulating SCI pathophysiology. So far, PDE4 subtype-specific therapies have not been tested in rodent SCI models. Despite this, PDE4B deficiency improved functional recovery starting 14 dpi until 42 dpi in a contusion SCI mouse model [80]. PDE4B is highly expressed in immune cells of the CNS, hence the improved functional outcomes could be attributed to diminished neuroinflammation. The effect of PDE8 and PDE7 inhibition in SCI is unknown to our knowledge. An overview of the effects of cAMP-specific inhibition on SCI outcomes and the potential mechanism is described in Table 2.

## 5. Limitations and Further Implications for SCI

Modulating cAMP levels by cAMP-specific PDEs showed promising results in orchestrating the initial inflammatory reactions and boosting endogenous regeneration in SCI. Nevertheless, the divergence between human and animal models is well known, and hence, clinical translation is often hampered. First of all, the majority of human SCI is on the cervical level and often consists of a combination of a contusion injury along with a compression or spinal cord hemisection due to fractured bone fragments. In contrast, mouse and rat SCI is frequently induced on the thoracic level to minimize severe complications such as respiratory problems [136]. Additionally, rodent models for SCI resemble only one type of spinal injury, most commonly a contusion injury, and cannot therefore fully recapitulate the complexity of the human case [136]. Furthermore, the pathophysiology differs not only between humans and rodents but also among rodent models. For instance, neutrophil infiltration in human SCI peaks 1–3 dpi and increases until 10 dpi, while neutrophil infiltration in mice and rats peaks at 12 h pi and 1 dpi, respectively [87,137,138]. In contrast, microglia and macrophage responses of rodents resemble the human SCI pathology [138]. Lastly, lymphocyte infiltration in mice and humans peaks 14 dpi, while in rats it is highest between 3 and 7 days pi [87,138]. Therefore, combining different types of injury and animal models provides a better insight into the clinical potential of new therapies.

Different studies have shown an improvement in functional outcomes with PDE inhibition alone, while other research has demonstrated that PDE inhibition is insufficient. Besides the rodent model used, the administration route, dose, and timing of cAMP-specific PDE inhibitors are important in deciding outcomes for SCI. Schaal et al. tested the i.v., s.c., and oral rolipram administration and the effect of the administration route on cellular protection. The i.v. and s.c. administration routes improved neuronal and oligodendrocyte survival and the preservation of myelinated axons. In contrast, oral rolipram administration could only improve cellular survival and showed little preservation of myelinated axons compared to the s.c. or i.v. route [139]. The route of administration determines the acute bioavailability in the spinal cord lesion, e.g., i.v. injection is known to have a rapid and sharp peak of the administered compound, while s.c. injection can provide a slower increase in bioavailability in the blood [139]. Both the i.v. and s.c. delivery routes demonstrated similar effects on cellular protection, an indication of a broad therapeutic window of rolipram, but the therapeutic windows of other PDE4, PDE7, or PDE8 inhibitors will be different. Besides the route of administration, the dose of the used PDE inhibitor is considered a crucial factor for determining outcomes. Data indicate that a low dose of 0.5 mg/kg/day s.c. rolipram administration for 14 days did not improve functional recovery in thoracic SCI in rats, while with the same route and timing of administration a significant improvement in functional recovery was observed with 1 mg/kg/day rolipram treatment. Furthermore, a higher dose of 0.8 mmol/kg/day s.c. was not able to improve functional outcomes or influence the pathophysiology of SCI, indicating that off-target effects or the influence on other PDE isoforms can abolish the beneficial effects of PDE4 inhibition in SCI [49,71,123]. Lastly, the timing of administration can influence the pathophysiology and hence outcomes. Rapid administration of PDE4 and PDE7 inhibitors after the initial injury showed the most beneficial outcomes on neuroinflammation, neuroprotection, and functional outcomes.

PDE4 inhibition is the most widely studied cAMP-specific modifying therapy in SCI, but its clinical translation is hampered mainly because of severe emetic side effects. Second-generation PDE4 inhibitors, including roflumilast, FCPR03, and FCPR16, show lower emetic potential [140,141,142]. Nevertheless, the molecular fingerprint of PDE4 subfamilies in the CNS provides the opportunity to affect neuroinflammation in the early phases and neuroregeneration in later phases of SCI to improve functional outcomes and limit side effects. Additionally, other cAMP-specific PDEs, such as PDE7, can be promising targets since they have been shown to modulate the immune cell function of resident and infiltrating immune cells. However, most studies on PDE7 have been performed in vitro, and hence, new research is needed to develop the inhibitory role of PDE7 in SCI recovery. Lastly, little is known about the effect of PDE8 inhibition on cellular responses, and more research is demanded to unravel its role in CNS pathology.

In conclusion, the immunomodulatory properties of PDE4, PDE7, and PDE8 are indicated to hamper the production of inflammatory cytokines and skew polarization toward repair-promoting neutrophils, phagocytes, and lymphocytes. Therefore, inhibition of cAMP-specific PDEs gives us the opportunity to tackle neuroinflammation in SCI and hence create a regenerative stimulating environment. Besides regulating neuroinflammation, PDE4 and PED7 have been shown to enhance neuronal survival directly and are postulated to promote neuroregeneration via inhibiting growth cone collapse. Additionally, indirect mechanisms imply that the immunomodulatory properties of cAMP-specific PDE inhibition create a spinal cord environment that promotes tissue repair. The immune and neuronal regulating properties of different PDE families can be attributed to specific genes, e.g., PDE4B was demonstrated to be a crucial subfamily in regulating phagocyte polarization. In conclusion, cAMP-specific PDE inhibition is considered a promising target for SCI therapy. A summary of the role of cAMP-specific PDEs in SCI is provided in Figure 1.

## Figures and Tables

**Figure 1 ijms-24-08135-f001:**
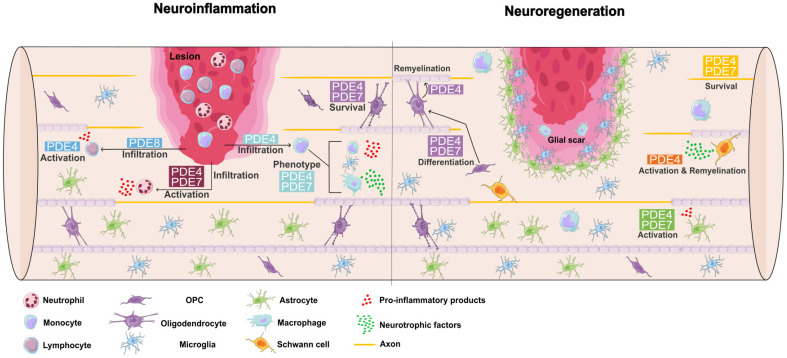
Overview of the involvement of cAMP-specific PDEs in cellular functioning during SCI. PDE4 is involved in regulating monocyte infiltration and PDE4, along with PDE7 in deciding the phenotype of infiltrated and resident phagocytes. Neutrophil infiltration and activity are regulated by PDE4 and PDE7, while lymphocyte infiltration is orchestrated by PDE8. PDE4 and PDE7 modulation can play a promising role in promoting OPC, oligodendrocyte and neuronal survival, along with PDE4 inhibition to stimulate remyelination by oligodendrocytes. The production of neurotrophic factors by Schwann cells and their myelination capacity can be increased upon PDE4 inhibition. Lastly, PDE4 and PDE7 were proven to play a role in astrocyte activation. OPC, oligodendrocyte precursor cell; PDE, phosphodiesterase. The figure is made with images from Servier Medical Art edited in Inkscape.

**Table 1 ijms-24-08135-t001:** Commonly used markers for fluorescent spinal cord stainings.

Cell Type	Marker
**Neuroinflammation**	
Neutrophils	MPO Anti-Ly6g Rhodamine 6G
Phagocytes	CD68 CD163 Iba1 CCR2 CD11b TREM119
Lymphocytes	CD5 (pan lymphocyte) CD4 (T-helper cells) CD8 (Cytotoxic T-cell) CD19 (pan B-cell) B220 (B cells)
**Neuroregeneration**	
Neurons	Neurofilament BDA βIII Tubulin NeuN 5HT
Oligodendrocytes	OLIG2 APC-CC1 MBP
Schwann cells	P0 Sox10 Nestin Sox2
Astrocytes	GFAP S100β

MPO, Myeloperoxidase; CD68, Cluster of Differentiation 68; CD163, Cluster of Differentiation 163; Iba1, Ionized calcium-binding adapter molecule 1; CCR2, C-C chemokine receptor 2; CD11b, Cluster of Differentiation 11b; TREM119, Triggering receptor expressed on myeloid cells 119; CD5, Cluster of Differentiation 5; CD4, Cluster of Differentiation 4; CD8, Cluster of Differentiation 8; CD19, Cluster of Differentiation 19; BDA, Biotinylated dextran amine; NeuN, Neuronal nuclear protein; 5HT, Serotonin; OLIG2, Oligodendrocyte transcription factor 2; APC-CC1, Anti-adenomatous polyposis coli clone CC1; MBP, Myelin basic protein; P0, Peripheral myelin; Sox10, SRY-related HMG-box 10; Sox2, SRY-related HMG-box 2; GFAP, Glial fibrillary acidic protein; S100β, S100 calcium-binding protein B.

**Table 2 ijms-24-08135-t002:** In vivo studies investigating the role of cAMP-specific PDE inhibition in SCI rodent models.

Author (Year)	Treatment Regime	SCI Model	Functional Outcomes	Proposed Mechanism
Beaumont et al., (2009) [123]	S.c. administration of 0.5 mg/kg/day PDE4 inhibitor rolipram for 2 weeks	C5-6 contusion SCI in female Sprague Dawley rats	↓ 25% hindlimb footfall error with rolipram treatment but not in forelimbs error in gridwalk test 5 weeks pi	No difference spared white matter 5 weeks pi in SCI lesion ↑ Oligodendrocyte somata in VLF 5 weeks pi with rolipram treatment
Bao et al., (2011) [49]	I.v. administration of 0.5 mg/kg PDE4 inhibitor IC486051 administered 2, 12, 24, 36, 48, and 60 h pi	T4 compression SCI female Wistar rats	↑ BBB locomotor score of 1.3 compared to vehicle, 8 weeks pi with IC486051 treatment ↓ Mechanical allodynia measured with Semmes-Weinstein filament test 6 weeks pi	↓ Number of phagocytes and neutrophils in the lesion 72 h pi ↓ Free radical formation in the spinal cord lesion
Schaal et al., (2012) [71]	I.v. administration of 1 mg/kg/day PDE4 inhibitor rolipram initiated 1 h pi, whereafter daily for 2 weeks	T8 contusion SCI in female Fischer rats	↑ BBB locomotor scores 6 weeks pi (control = 11.1 vs. roflumilast = 12.9 rolipram)	↑ Neuronal and oligodendrocyte survival in lesion ↓ Number of phagocytes 2 weeks pi in lesion
Costa et al., (2013) [134]	S.c. administration of 3.18 mg/kg/day PDE4 inhibitor rolipram for 2 weeks	T10 contusion SCI in female Wistar rats	↑ BBB locomotor scores (control = 11.7 vs. rolipram = 14.6) 8 weeks pi ↑ Locomotor outcomes on beamwalk test (control = 2.5 vs. rolipram = 4.5) 8 weeks pi	↑ White matter sparing in lesion No difference lesion volume and lesion length
Myers et al., (2019) [80]	No treatment	T9 contusion SCI in C57BL/6 PDE4B KO mice	↑ BMS locomotor score of 1.3 compared to vehicle 42 dpi in PDE4B KO mice ↑ Locomotion tread-mill scan analysis	↓ Inflammatory markers of macrophages and astrocytes in lesion ↑ Oligodendrocyte maturation and white matter sparing 42 dpi in lesion
Moradi et al., (2020) [78]	I.p. administration of 0.5 or 1 mg/kg PDE4 inhibitor roflumilast 30 min before SCI induction	T9 contusion SCI in male rats	↑ BBB locomotor score (control = 9 vs. roflumilast = 12) 28 dpi ↑ Mechanical allodynia 28 dpi ↓ Thermal sensitivity 28 dpi	↓ Cavity size ↑ Anti-inflammatory responses in lesion 28 dpi

S.c., subcutaneous; PDE, phosphodiesterase; pi, post injury; SCI, spinal cord injury; ↓, reduced; ↑, elevated; VLF, ventrolateral funiculus; i.v., intravenous; BBB, Basso, Beattie and Bresnahan; BMS, Basso Mouse Scale; i.p., intraperitoneal.

## Data Availability

Not applicable.

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
