# Peer review of "Unleashing Spinal Cord Repair: The Role of cAMP-Specific PDE Inhibition in Attenuating Neuroinflammation and Boosting Regeneration after Traumatic Spinal Cord Injury"

_ijms, 2023, doi:10.3390/ijms24098135_

Round 1

Reviewer 1 Report

The overall logic of this review is good. Although it illustrates the situation of nerve inflammation, nerve regeneration and functional recovery, some tissue fluorescence sections of corresponding spinal cord cross-section can be appropriately added.

Some English grammar needs to be re-edited.

Author Response

We thank the reviewer for carefully reading our review. As requested, the entire manuscript has been re-edited for the English language using the Grammarly software. Although the reviewer suggests adding fluorescent spinal cord stainings to the manuscript, we do not prefer to display images that have been published in other papers. In addition, a new paper regarding the role of PDE4 in spinal cord injury, including fluorescent spinal cord sections, is being finalized and will refer to this review. However, we have added a table with commonly used markers for assessing neuroinflammation and neuroregeneration using fluorescent stainings of spinal cord tissue (Table 1). We sincerely hope the reviewer agrees with our suggestion.      

Reviewer 2 Report

This article by Femke Mussen et al reviews the potential therapeutic role of inhibiting the enzyme phosphodiesterases (PDE) (involved in the hydrolysis of cAMP in cells) in traumatic spinal cord injury. The review is written very well and nicely introduces the concept of severe neuroinflammation that is often accompanied by SCI and later hampers the regeneration and functional repair process. The review nicely describes the role of the different PDE isoforms in neuroinflammation following SCI and in neuroregeneration following SCI in specific subtypes. The language is professional and acceptable for publication in IJMS.

Author Response

We thank the reviewer for these kind comments.